# Effects of Sodium Formate and Calcium Propionate Additives on the Fermentation Quality and Microbial Community of Wet Brewers Grains after Short-Term Storage

**DOI:** 10.3390/ani10091608

**Published:** 2020-09-09

**Authors:** Jingyi Lv, Xinpeng Fang, Guanzhi Feng, Guangning Zhang, Chao Zhao, Yonggen Zhang, Yang Li

**Affiliations:** College of Animal Science and Technology, Northeast Agricultural University, Harbin 150030, China; 18846440487@163.com (J.L.); dnfangxinpeng@163.com (X.F.); fengguanzhi123@163.com (G.F.); zgn1234@126.com (G.Z.); hljsyszhaochao@163.com (C.Z.); zhangyonggen@sina.com (Y.Z.)

**Keywords:** wet brewers grains, sodium formate, calcium propionate, fermentation quality, microbial community

## Abstract

**Simple Summary:**

The objective of this study was to examine the effect of sodium formate and calcium propionate on the fermentation quality and microbial community of wet brewers grains (WBG) after short-term storage. Both additives improved the silage quality of WBG ensiled for 20 days to different extents. However, ensiled WBG treated with sodium formate had higher contents of dry matter, water-soluble carbohydrates, and neutral detergent fibers and better fermentation quality, rumen degradation, and microbial composition. The addition of sodium formate enhances the abundance of desirable *Lactobacillus* and reduces the abundance of undesirable microorganisms, including *Clostridium*. In summary, during short-term storage of high-moisture feed, sodium formate has a more beneficial preservation effect than an equivalent dose of calcium propionate.

**Abstract:**

The objective of this research was to examine the effect of sodium formate (SF) and calcium propionate (CAP) on the fermentation characteristics and microbial community of wet brewers grains (WBG) after short-term storage. In the laboratory environment, fresh WBG was ensiled with (1) no additive (CON), (2) sodium formate (SF, 3 g/kg fresh weight), and (3) calcium propionate (CAP, 3 g/kg fresh weight) for 20 days. After opening, fermentation characteristics, chemical composition, rumen effective degradability, and the microbial community of ensiled WBG were analyzed. The addition of CAP had no effect on pH and lactic acid concentration and increased the concentrations of propionic acid; the SF group had the lowest pH and acetic acid, butyric acid, and ammonia nitrogen contents and the highest lactic acid concentration. After fermentation, the SF group had the highest contents of dry matter (DM), water-soluble carbohydrates (WSCs), and neutral detergent fiber (NDF). The contents of the three nutrients in the CAP group were significantly higher than those in the CON group. The addition of the two additives had little influence on the crude protein (CP) and acid detergent fiber (ADF) contents of the ensiled WBG. Two additives elevated in situ effective degradability of DM and NDF compared with the parameters detected in the CON group; WBG ensiled with SF had higher effective in situ CP degradability than that in the CON and CAP groups. The results of the principal component analysis indicate that the SF group and two other groups had notable differences in bacterial composition. The analysis of the genus level of the bacterial flora showed that the content of *Lactobacillus* in the SF group was significantly higher than that in the two other treatment groups, while the content of *Clostridium* was significantly lower than that in the two other treatment groups. Therefore, the addition of sodium formate can suppress the undesirable microorganisms, improve the fermentation qualities, and ensure that WBG is well preserved after 20 days of ensiling.

## 1. Introduction

Wet brewers grain (WBG) is a byproduct of the brewing process and is a suitable alternative feed source for dairy [1] and beef cattle [2,3] due to its unique nutrient composition combined with low price. However, the seasonality and low dry matter (DM) content, which hinders storage and preservation of WBG, are the main limiting factors for the effective use of WBG [4]. Therefore, it is very important to determine an appropriate preservation strategy according to the characteristics of WBG.

Silage is a common processing method of high-moisture feed; however, most sugars are removed from WBG during the malting process leaving inadequate levels of the substrate (water-soluble carbohydrates, WSCs) for silage fermentation, which can result in silage failure [5]. Adding a substrate to WBG to enhance its DM and WSC concentrations is an option that can improve the fermentation effect of WBG. However, reports in the literature are contradictory. Compared with WBG alone, WBG mixed with beet pulp pellets [6] and dry group corn [5] improved fermentation. In contrast, compared with WBG alone, the addition of soy hulls to WBG increased DM and nutrient losses [7]. Furthermore, in actual production, it is difficult to ensure even mixing of a large amount of WBG and a substrate. After comprehensive consideration of these factors, in this experiment, we used inhibitors of harmful bacteria to achieve short-term preservation of WBG.

Organic acid additives, especially short-chain fatty acids, can directly acidify the feed with low WSC concentration and high buffering capacity, which can immediately reduce pH, inhibit the activity of undesirable microorganisms, and eventually reduce the nutrient loss of the crop [8,9]. Formic acid and propionic acid were compared with other short-chain fatty acids and have been considered as silage additives in numerous studies [8,10]. Formic acid is the most acidic of organic acids and is a fermentation inhibitor; the advantage of formic acid is its suitability for the storage of high-moisture raw materials due to a reduction in the loss of dry matter and nutrients to ensure silage quality [8]. Propionic acid is the weakest of organic acids that has the strongest antifungal activity and is an aerobic spoilage inhibitor [11,12]. However, due to the corrosiveness and harmful properties of formic acid and propionic acid, their applications pose certain problems. Thus, their salt derivatives have been developed; these derivatives are considered safer and easier to handle. The effects of sodium formate (SF) and calcium propionate (CAP), the salts of formic acid and propionic acid, respectively, are identical to the effects of the corresponding acids after ionization in water [13,14]. SF has become a commercial silage additive and is almost as effective as formic acid. During the silage fermentation process, formic acid can be quickly released, thereby rapidly reducing the pH (pH < 4.2) and inhibiting the growth of undesirable microorganisms [15], *Clostridium* in particular [13], while increasing the lactic acid content [16]. Previous studies have claimed that propionic acid has potent antifungal properties and that its application at the level of 0.2–0.3% of fresh forage weight has a positive effect on the stability of corn silage [17,18]. However, the acidity of organic acids decreases as the chain length increases; hence, a decrease in the pH value of silage in the presence of CAP is slower than that observed in the case of silage treated with SF, and pH in the CAP silage is higher than that in the formic acid silage [8]; this property may influence the short-term preservation effect in high-moisture feed. Ensiling is the process of microbial action, and bacteria play an essential role during ensiling. Using reasonable detection methods to assess the bacterial community of SF and CAP after short-term ensiling is very important because it can determine the mechanism of action of the two additives in the preservation of WBG. The silage microbiota has been quantified and analyzed by next-generation sequencing [19,20]. Hence, the target of our study was to investigate fermentation quality, rumen degradation, and the microbial community of WBG supplemented with SF and CAP for a short time and to clarify the preservation effect of two additives used at equivalent doses.

## 2. Materials and Methods

### 2.1. Silage Preparation and Treatments

Brewers grains used in the present study were provided free by Jinrun Animal Husbandry Company Limited (Harbin, China). Fresh WBG was treated with (1) no additive (CON), (2) sodium formate (SF; 98%, Chenyuan Fine Chemical Co., Ltd., Jiangyan, China) applied at 3 g/kg fresh weight (FW) [13], and (3) calcium propionate (CAP; 98%, Chenyuan Fine Chemical Co., Ltd., Jiangyan, China) applied at 3 g/kg fresh weight [10]. The two additives were dissolved in deionized water and applied as a solution. Additives were homogenously mixed into WBG using a hand sprayer. After proper mixing, the treated WBG was packed into a 20 × 30 cm plastic laboratory fermentation bag (Chuangjia Packaging Material Co., Ltd., Wenzhou, China) and the air was subsequently removed by a vacuum packing machine (Maige Automation Equipment Co., Ltd., Qingdao, China). Thirty-six bags (1 ensiling day × 3 treatments × 12 repeats) were prepared and incubated indoors at ambient temperature (20 ± 2 °C) for 20 days. Twelve bags for each treatment were opened to analyze fermentation characteristics, chemical compositions, in situ effective degradability, and microbial community.

### 2.2. Analysis of Chemical Composition, Fermentation Characteristics, and In Situ Effective Degradability

Prior to fermentation, 10 g of fresh WBG was homogenized in 90 mL of sterilized saline water for 2 h, and the homogenate was serially diluted from 10–1 to 10–5 in sterilized water. The samples were cultured in a 37 °C anaerobic incubator for 48 h and the lactic acid bacteria (LAB) were counted on the de Man Rogosa Sharp agar (MRS) medium (Shanghai Bio-way Technology Co., Ltd., Shanghai, China). Potato dextrose agar (Nissui) was used to count the mold and yeast after growth at 30 °C for 24 h; yeasts were distinguished from molds and other bacteria by observing the appearance and cell morphology of the colonies. The colonies were counted and recalculated as a viable number of microorganisms (cfu/g FM-1) to use in statistical analysis.

After 20 days of ensiling, silage samples were homogenized. Ten grams of the subsample was mixed with 90 mL deionized water and placed at 4 °C for 24 h; then, the flowing samples were filtered through 4 layers of cheesecloth. A portable pH meter (Sartorius basic pH meter, Göttingen, Germany) was immediately used to measure the pH of the extract. An aliquot of the silage extract was centrifuged at 1800× *g* for 15 min at 4 °C and organic acids (lactic acid, acetic acid, propionic acid, and butyric acid) and ammonia-N were analyzed. The phenol/hypochlorite method was used to determine ammonia-N [21]. Organic acids in the filtrate were quantified by high-performance liquid chromatography (Carbomix H-NP5 column, 55 °C, 2.5 mM H2SO4, 0.5 mL/min) according to the method of Yuan et al. [22].

Fresh WBG and the subsamples were dried at 65 °C for 48 h and ground with a high-speed universal mill to pass through a 1-mm sieve grind. The DM and crude protein (CP) contents were determined according to the procedures of the (Association of Official Analytical Chemists, AOAC) [23]. Acid detergent fiber (ADF) and neutral detergent fiber (NDF) concentrations were analyzed using an Ankom 220 fiber analyzer (Ankom Technology Corp., Macedon, NY, USA) according to the method of Van Soest et al. [24]. Heat-stable α-amylase and sodium sulfite were not used due to the low starch content of WBG. The WSC contents were determined by colorimetry after reaction with anthrone reagent [25].

Three ruminally cannulated Holstein cows were housed to determine the in situ rumen degradation of DM, CP, and NDF of ensiled WBG. The basal diet of the Holstein cows (g/kg of DM) consisted of 102 g/kg Chinese wild rye hay, 180 g/kg alfalfa hay, 215 g/kg corn silage, and 503 g/kg concentrate mixture; the cows were fed twice daily for a total dry matter intake (DMI) of 15 g/kg of BW. This study was performed in strict accordance with the recommendations of the National Research Council Guide, and all animal experimental procedures were approved by the Northeast Agricultural University Animal Science and Technology College Animal Care and Use Committee (Protocol number: NEAU-[2011]-9). The method of Nuez-Ortín and Yu was used to measure the in situ rumen degradation of ensiled WBG [26]. All bags were incubated in the rumen for 48, 36, 24, 16, 12, 8, 4, and 0 h according to the ‘gradual addition/all out’ schedule. The constants and the in situ effective degradability were estimated based on the nonlinear model according to the method described by Hao et al. [27].

### 2.3. Microbial Diversity Analysis

#### 2.3.1. DNA Extraction

Ten grams of fresh samples from the CON, SF, and CAP groups fermented for 20 days were mixed with 90 mL of aseptic 0.85% NaCl solution. The solution was shaken violently at the speed of 120 r/min for 2 h and filtered through 4 layers of gauze and the filtrate was centrifuged for 10 min at 4 °C at the speed of 10,000 r/m. After centrifugation, the supernatant was removed, and the sediment was suspended in 1 mL of aseptic 0.85% NaCl solution. The microbial particles were obtained by centrifugation for 10 min at 4 °C. The microbial DNA was extracted by Fast DNA SPIN extraction kits (MP Biomedicals, Santa Ana, CA, USA) according to the manufacturer’s protocols and quantified using a NanoDrop ND-1000 spectrophotometer (Thermo Fisher Scientific, Waltham, MA, USA).

#### 2.3.2. PCR Amplification and High-Throughput Sequencing of Metagenomic DNA

The V3-V4 region of bacterial 16S rRNA genes was amplified by PCR with forward primer 338F (5′-ACTCCTACGGGAGGCAGCA-3′) and reverse primer 806R (5′-GGACTACHVGGGTWTCTAAT-3′). An aliquot of approximately 30 ng DNA of each sample was used for amplification. The thermal cycling included initial denaturation at 98 °C for 2 min followed by 25 cycles of denaturation at 98 °C for 15 s, annealing at 55 °C for 30 s, and extension at 72 °C for 30 s with final extension at 72 °C for 5 min. The PCR amplification products were purified with Agencourt AMPure beads (Beckman Coulter, Indianapolis, IN, USA) and quantified by a PicoGreen dsDNA assay kit (Invitrogen, Carlsbad, CA, USA).

After purification and quantification of the PCR products, the DNA fragments of the community were sequenced by double-terminal (paired-end) sequencing using an Illumina MiSeq platform (Wuhan Frasergen Bioinformatics Co., Ltd., Wuhan, China). The barcodes and primers were excluded to obtain high-quality sequencing results. Sequencing data were processed using the Quantitative Insights Into Microbial Ecology (QIIME, v1.8.0 http://qiime.org/) pipeline as described previously [28]. Beta diversity analysis was performed to investigate the structural variation of microbial communities across the samples using UniFrac distance metrics and the results were visualized via the principal coordinate analysis [29,30]. Principal component analysis (PCA) was conducted based on the genus-level compositional profiles [31]. Taxa abundance at the genus level were statistically compared between the samples or groups by Metastats [32] and visualized as violin plots. To reveal the variation of microbial communities in the groups, LEfSe (linear default parameters) was used as a monitoring model [33].

### 2.4. Statistical Analyses

The effects of SF and CAP on fermentation indicators, chemical composition, and in situ effective degradability of ensiled WBG were evaluated using the GLM procedure of SAS software system (version 9.4; SAS Institute Inc., Cary, NC, USA); the differences were considered significant at *p* < 0.05. The correlation between microbial abundance and fermentation index was tested by the Pearson correlation analysis (*p* < 0.01 for extremely significant correlation and *p* < 0.05 for significant correlation).

## 3. Results

### 3.1. Wet Brewers Grain Characteristics

Table 1 shows the chemical composition and microbial populations of WBG. WBG had a DM of 27.28% FW, and contents of NDF, ADF, CP, and WSC were 67.94%, 20.53%, 26.95%, and 0.53% DM, respectively. WBG contained 4.75 log10cfu/g FW of LAB and 3.79 log10cfu/g FW of yeasts; mold was not detected in WBG.

### 3.2. Fermentation Characteristics, Chemical Composition, and In Situ Effective Degradability of SF- and CAP-Ensiled WBG after 20 Days

The fermentation characteristics, chemical compositions, and in situ effective degradability of ensiled WBG after 20 days are shown in Table 2. Addition of SF significantly decreased the pH (*p* = 0.0068) and concentrations of ammonia-N (*p* < 0.0001), acetic acid (*p* < 0.0001), and butyric acid compared with the corresponding values in the CON and CAP groups. The contents of ammonia-N, acetic acid, and butyric acid in the CAP group were lower than those in the CON group. We detected propionic acid only in the CAP group.

A significant (*p* < 0.05) effect of SF and CAP addition on the DM, NDF, and WSC contents of the WBG silages was observed, whereas the additives did not influence the CP (*p* = 0.53) and ADF (*p* = 0.92) contents of the WBG silage. The addition of CAP improved the DM (*p* < 0.0001), NDF (*p* = 0.0003), and WSC (*p* = 0.0002) contents of WBG silages compared with the parameters detected in the CON group; WBG ensiled with SF had the highest contents of these compounds.

The addition of two additives significantly increased the in situ effective degradability of DM (*p* = 0.0039) and NDF (*p* = 0.0017) compared with that in the control group, and the values were the highest in WBG ensiled with SF. An increase in in situ effective CP degradability was found only in WBG ensiled with SF (*p* = 0.0111).

### 3.3. Microbial Community of Ensiled WBG

The composition of the bacterial community was compared between various treatments and analyzed by the β-diversity analysis. As shown in Figure 1, the principal component analysis clearly indicated variability of the microbial community. Separate analysis of the SF group samples versus the samples of the CON and CAP groups indicates that the addition of SF has an impact on the microbial community. Interestingly, the samples of the CAP group had a similar microbial community to that in the CON group.

The effects of the additives on the microbial community were additionally illustrated by the relative abundance of the genus-level bacterial community in Figure 2 and Table 3. *Lactobacillus* was the most predominant genus of all genera in the three groups (Figure 2). Figure 3 indicates that the improvement in the fermentation quality of the WBG silage may be due to the addition of SF to inhibit undesirable microorganisms, such as *Clostridium*, and to promote beneficial microorganisms, such as *Lactobacillus*. The contents of other desirable LAB, including mainly *Bacillus*, *Bifidobacterium*, and *Weissella*, were higher in WBG ensiled with SF. The dominant genera were *Lactobacillus* (84.6%), *Bacillus* (4.53%), and *Weissella* (4.07%) in WBG ensiled with SF; however, *Lactobacillus* (40.2%, 51.9%), *Clostridium* (51.31%, 31.14%), and *Prevotella* (3.58%, 7.16%) were the dominant microbes in the CON and CAP group samples, respectively (Figure 2).

### 3.4. Correlations between Relative Abundance of Bacteria and Fermentation Quality Indices

The correlations between the relative abundance of bacterial genera and silage fermentation indices are presented in Table 4. Silage pH was negatively correlated with the relative abundance of *Lactobacillus* (r = −0.73, *p* = 0.025), *Bacillus* (r = −0.85, *p* = 0.04), and *Weissella* (r = −0.92, *p* = 0.001) and positively correlated with the relative abundance of *Clostridium sensu stricto* 1 (r = 0.75, *p* = 0.02), 11 (r = 0.73, *p* = 0.025), and 12 (r = 0.92, *p* = 0.001). Lactic acid concentration positively correlated with the relative abundance of *Lactobacillus* (r = 0.77, *p* = 0.02) and *Bacillus* (r = 0.68, *p* = 0.042) and negatively correlated with the relative abundance of *Clostridium sensu stricto* 1 (r = −0.76, *p* = 0.02) and 11 (r = −0.80, *p* = 0.01). Acetic acid concentration negatively correlated with the relative abundance of *Lactobacillus* (r = −0.80, *p* = 0.01), *Bacillus* (r = −0.87, *p* = 0.003), and *Weissella* (r = −0.77, *p* = 0.02) and positively correlated with the relative abundance of *Clostridium sensu stricto* 11 (r = 0.85, *p* = 0.004) and 1 (r = 0.84, *p* = 0.005). The relative abundance of *Bifidobacterium* (r = 0.78, *p* = 0.01) and *Clostridium sensu stricto* 16 (r = 0.82, *p* = 0.007) was positively correlated with the propionic acid concentration. Butyric acid and ammonia-N concentrations positively correlated with the relative abundance of *Clostridium sensu stricto* 11 (r = 0.90, *p* = 0.001; r = 0.85, *p* = 0.004, respectively) and 1 (r = 0.87, *p* = 0.002; r = 0.90, *p* = 0.001, respectively) and negatively correlated with the relative abundance of *Lactobacillus* (r = −0.83, *p* = 0.006; r = −0.85, *p* = 0.003, respectively), *Bacillus* (r = −0.92, *p* = 0.001; r = −0.93, *p* = 0.0003, respectively), and *Weissella* (r = −0.81, *p* = 0.008; r = −0.83, *p* = 0.006, respectively).

## 4. Discussion

### 4.1. Chemical Composition and Microbial Population of Wet Brewers Grains before Ensiling

Due to its good nutritional composition and low price, WBG has been a common feed material for beef cattle and dairy cows. However, the very low DM content of WBG makes it difficult to preserve. Studies have shown that deterioration occurs promptly within 2 days after the silo opening if WBG was the only ensiled component [34]. The realization of short-term storage of WBG through simple, convenient, and economic means is of great significance for the feeding, storage, and transportation of this high-quality feed. The nutrient composition of WBG was 67.94%, 20.53%, and 26.95% of NDF, ADF, and CP, respectively. These concentrations are similar to the results reported by Moriel et al. [35]. The DM differences between WBG reported in different literature sources are quite large [5,7,35], and this variation is expected and related to the differences in the amount and type of grain used during the brewing process. However, numerous articles have reported that most of the sugar and starch of WBG are removed after malting and mashing processes [36], and the high water content makes it difficult for traditional microorganisms to ferment and preserve WBG. Moreover, the addition of a substrate to WBG has inconsistent fermentation effects on WBG and is not easy to achieve in production. These factors are the main reasons why we used organic acid salts to separately ferment WBG in the present study.

### 4.2. The Effects of Two Additives on Fermentation Quality, Chemical Composition, and Rumen Degradation of Wet Brewers Grain Silage

The pH of the silage is a traditional and effective indicator for evaluating fermented feed. The quality of the silage is ensured only when the pH of the silage is below 4.2 [8]. In addition to a low pH value, a high lactic acid content and a low ammonia-N and butyric acid content are considered to be important indicators of excellent silage quality [37]. In this study, the pH in the SF group decreased to 4.11 after 20 days of ensiling, while in the control and CAP groups, the pH was 4.44 and 4.40, respectively. This phenomenon may be related to the acidic properties of the additives. In this study, the acidity of the additives may have reduced the pH value of the SF group to 4.11 after 20 days of ensiling, while in the control and CAP groups, the pH was 4.44 and 4.40, respectively. Formate can quickly release formic acid during the silage fermentation process, thereby quickly reducing the pH and inhibiting the growth of undesirable microorganisms [38]. However, a decrease in the pH value in the CAP group was slower than that in the SF group because the acidity of the organic acid was decreased in the case of a longer chain length. This discovery is consistent with Woolford, who pointed out that formic acid has a lower dissociation constant Ka, which is the reason why formic acid can induce a more pronounced significant reduction in the pH value of the culture medium and feed crops [39]. Therefore, the lower pH in the SF group inhibited the growth of harmful bacteria and production of butyric acid, thus causing this group to have the lowest levels of ammonia-N and butyric acid. Rapid acidification reduces the risk of early undesirable microorganism growth; the rate of a decline in pH is as essential as the final pH. Moreover, formic acid has a significant antibacterial effect, which can inhibit the growth and reproduction of clostridia and some undesirable bacteria. Additionally, the respiration of the silage raw materials can be suppressed to enable the production of high-quality silage from the silage raw materials even if the WSC content is not high. This feature makes formic acid more suitable for the storage of low dry matter and low sugar content raw materials [16]. CAP plays a role in the preservation of fresh alfalfa; however, we found that CAP did not effectively reduce the pH of the fermented feed according to the literature data [8,40]. This feature may explain why the CAP group did not have good fermentation properties in our experiment.

The main purpose of the silage is to reduce the loss of feed nutrients and dry matter; effluents and gases are the main factors leading to the DM loss during silage [8]. In our experiment, the DM content of the SF group was the highest, which is related to the activity of LAB during the fermentation of the feed [41]. However, the loss caused by carbon dioxide can significantly increase when the heterofermentative bacteria, as *Clostridium*, *Enterobacteria*, and yeasts facilitate fermentation. In the control and CAP groups, we detected a higher content of butyric acid, indicating that the fermentation by *Clostridia* reduces the quality of the feed. The proliferation of harmful microorganisms also decomposes protein, sugar, and hemicellulose, which significantly increase the ammonia-N concentrations in these groups, while the loss of DM, NDF, and WSC content is increased. *Clostridia* growth develops during the early stages of ensiling without inhibitor supplementation, especially in feeds with a high buffer ability and low content of WSC [42]. Therefore, the faster acid production rate of SF compared to that of CAP has an advantage in WBG fermentation, thus ultimately influencing the fermentation effects of these two additives. In our experiment, SF significantly reduced the loss of DM, which may be due to the antibacterial property of SF against harmful bacteria and the ability of SF to rapidly reduce pH. Moreover, higher loss of DM is also related to the concentration of acetic acid [43]. Goeser et al. demonstrated that there is a positive correlation between the acetic acid concentrations and the loss of DM in silage and that the activity of acetic acid-producing bacteria can exacerbate the loss of DM [44]. In our experiment, we were pleasantly surprised to find that the SF group had the lowest acetic acid content and the highest lactic acid content. Studies by other authors have confirmed that adding formic acid to the silage can reduce its pH (pH < 4.2), increase the concentration of lactic acid, and inhibit the growth of harmful microorganisms [16]. The bacteriostatic effect of formic acid reduces the activity and lactic acid content of LAB in the early stage of fermentation; however, LAB can continue growing in the presence of formic acid for a certain period of time. The significant increase in propionic acid content in the CAP group is consistent with an increase observed by adding CAP before ensiling. Propionic acid exhibits strong antifungal properties and has a positive effect on the stability of the corn silage [10], which may be due to the weak lowering of pH of the feed; hence, CAP did not show a good fermentation effect in this experiment. It is also possible that propionic acid is not suitable for the preservation of the feeds with a higher water content.

In situ effective degradability measurements are routinely used to determine the nutritive values of ruminant feeds [26]. The DM rumen degradation rate is an important factor that influences the DMI of dairy cows and is used to assess the nutritional value of a feed. This parameter can reflect the degree of feed utilization in the rumen and mainly includes the degradation of carbohydrates, proteins, and other substances. In this study, the addition of SF and CAP increased the in situ effective degradability of DM and CP apparently due to a reduction in the losses of the WBG silages treated with additives and consequently provide more available substrates for microbial degradation in the rumen [37]. Addition of SF increased the in situ effective degradability of NDF compared with that in the control and CAP groups; this result may be due to better fermentation effects of SF, which reduce the loss of hemicellulose and other components in NDF caused by undesirable microorganisms. Thus, the content of hemicellulose in WBG of the SF group was higher than that in the control and CAP groups. Hemicellulose is an easier fermentable fiber, and its content will directly influence the rate and extent of NDF degradation in the rumen [10]. SF is a better preservative of the nutrients in WBG and has a significant effect on the improvement of feed application after short-term silage.

### 4.3. The Effects of Two Additives on the Bacterial Community of Wet Brewers Grain Silage

Next-generation sequencing provides more detailed diagrams of bacterial communities than traditional sequencing techniques, thus helping to demonstrate the response of bacterial communities to the state of the silage [45]. In this study, the variability in the microbial community was clearly detected by the result of the principal coordinate analysis (Figure 1). Analysis of the samples of WBG fermented alone or in combination with SF for 20 days demonstrated that SF has a distinct effect on the microbial community. The differences in the bacterial community induced by the addition of CAP to WBG silage were also very clear. Studies using next-generation sequencing in fermented WBG are rare; however, the changes in the microbial community may account for the diversity in the silage quality [46].

The analysis of Figure 3 and Table 4 indicates that the dominant genus of WBG silage differed depending on the treatments. *Lactobacillus* was the predominant microbe in WBG ensiled with SF, while the samples of the control and CAP groups primarily included *Lactobacillus*, *Clostridium*, and *Prevotella*. Normally, aerobic microbes deplete oxygen and anaerobes, such as *Lactobacillus*, develop and reduce the pH of the silage during the early phase of fermentation. In this study, the addition of SF rapidly reduced pH, inhibited other undesirable microorganisms, and inhibited the activity of *Lactobacillus*; however, extension of the fermentation time resulted in gradual recovery of *Lactobacillus* activity and vigor so that it occupied a dominant position in ensiled WBG [47], while other undesirable microorganisms were completely suppressed. Interestingly, the richness of *Weissella* in the SF group was also at a higher level compared with that in the control and CAP groups. The species pertaining to *Weissella* are obligate heterofermentative bacteria, which are often replaced by acid-tolerant *Lactobacillus* in a later stage of the silage [48]; the reason for this phenomenon is unclear. The reason for SF-induced enhancement of *Bacillus* in ensiled WBG is also unclear; however, reports in the literature indicate that the majority of *Bacillus* are harmless and that the antibacterial substances produced by microorganisms have broad-spectrum bactericidal activity; thus, *Bacillus* facilitates the generation of lactic acid [49,50]. *Clostridium* is deemed to be an undesirable microorganism in silage due to its potential to cause excessive protein degradation, DM loss, and butyric acid production, thereby promoting the growth of less acid-resistant putrid microorganisms and thus resulting in reduced silage intake. The addition of CAP inhibited the number of *Clostridium* to an extent; however, CAP did not effectively reduce pH to inhibit the growth of harmful bacteria. Hence, the undesirable microorganisms, such as *Clostridium*, may compete with *Lactobacillus* for nutrients and generate ammonia-N. The data of Figure 1 indicate that the samples of the CAP group are similar to the control group and that the microbial community of the CAP group is similar to that of the control group. Moreover, in the control and the CAP groups, an increase in the content of *Prevotella*, which consumes more protein, may be a reason for the production of higher levels of acetic acid.

The results shown in Table 4 indicate that there is an interaction between the bacterial community and silage fermentation. Spearman correlation analysis indicates that pH, lactic acid, acetic acid, propionic acid, butyric acid, and ammonia-N are the principal factors that determine the composition and diversity of the bacterial community. In this study, we found that the relative abundance of *Lactobacillus* was negatively correlated with pH, acetic acid, butyric acid, and ammonia-N and positively correlated with lactic acid in the silage; these results are similar to the data of a previous study [48]. Our study found a few bacterial communities that are sensitive to additives. The changes in the dominant group may be related to physiological characteristics and adaptation of the main group of bacteria to environmental changes caused by the additive treatment. SF and CAP can influence bacterial physiology directly and the microbial community indirectly. The silage fermentation time was only 20 days in our study; however, the overall trend reflects the impact of various additives on the preservation of WBG.

## 5. Conclusions

SF and CAP additives can enhance the quality of WBG silage to variable degrees; however, at the same dose, ensiled WBG treated with SF had a better fermentation quality, rumen degradation, and microbial composition. The addition of SF increased the abundance of desired *Lactobacillus* and decreased the abundance of undesirable microorganisms, such as *Clostridium*. In conclusion, SF has a better preservation effect in short-term storage of high-moisture feed than an equivalent dose of calcium propionate. In production, the addition of SF can prolong the storage time of WBG, thereby improving the operability of using of WBG on dairy farms.

## Figures and Tables

**Figure 1 animals-10-01608-f001:**
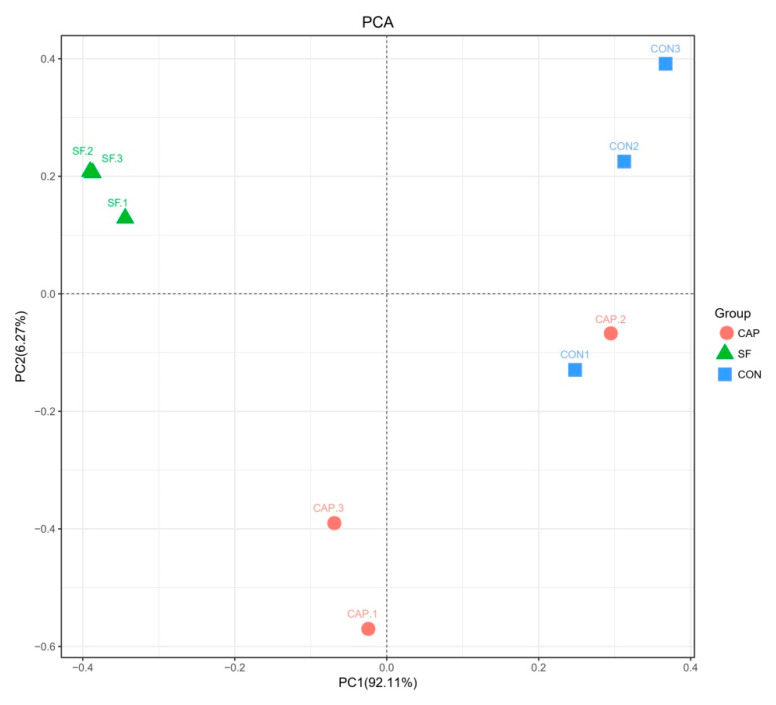
Principal component analysis of the bacterial community in wet brewers grains ensiled with sodium formate and calcium propionate after 20 days (CON, control group; SF, sodium formate; CAP, calcium propionate; 1, 2, 3, three replicates for each treatment).

**Figure 2 animals-10-01608-f002:**
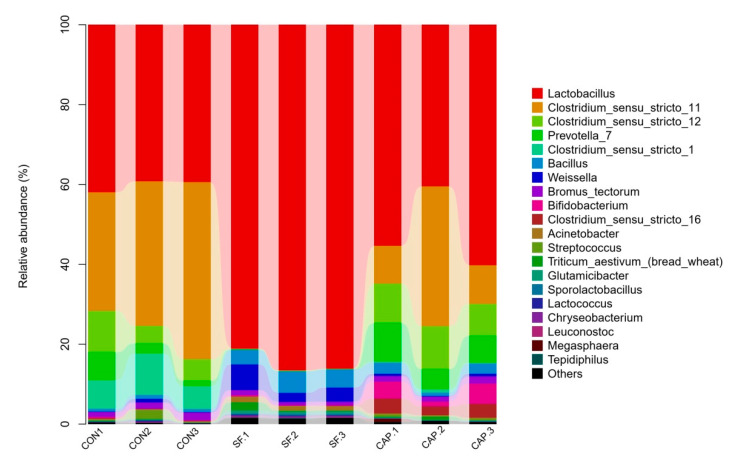
Bacterial community and relative abundance by genus of wet brewers grains ensiled with sodium formate and calcium propionate after 20 days (CON, control group; SF, sodium formate; CAP, calcium propionate; 1, 2, 3, three replicates for each treatment).

**Figure 3 animals-10-01608-f003:**
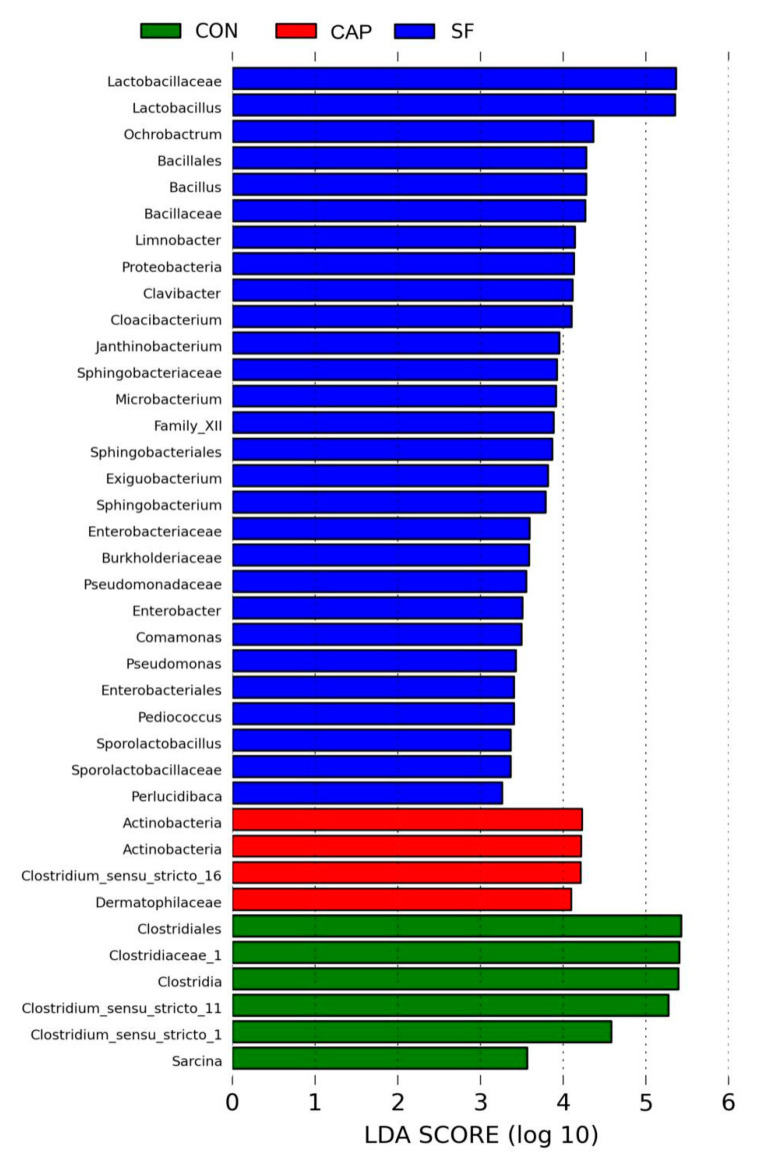
LEfse (Linear discriminant analysis Effect Size.) online tool was used to compare the microbial changes in wet brewers grains ensiled with sodium formate and calcium propionate after 20 days (CON, control group; SF, sodium formate; CAP, calcium propionate).

**Table 1 animals-10-01608-t001:** Chemical compositions and microbial populations of wet brewers grains before ensiling.

Items	Mean ± SD
DM, % FW	27.82 ± 1.23
NDF, % DM	67.94 ± 2.14
ADF, % DM	20.53 ± 1.12
CP, % DM	26.95 ± 1.46
WSC, % DM	0.53 ± 0.02
LAB, log_10_cfu/g FW	4.57 ± 0.013
Yeast, log_10_cfu/g FW	3.79 ± 0.015
Mold, log_10_cfu/g FW	ND

ADF, acid detergent fiber; CFU, colony forming units; CP, crude protein; DM, dry matter; FW, fresh weight; LAB, lactic acid bacteria; ND, not detected; NDF, neutral detergent fiber; SD, standard deviation; WSC, water-soluble carbohydrates.

**Table 2 animals-10-01608-t002:** Fermentation characteristics, chemical compositions, and in situ effective degradability of ensiled wet brewers grains for 20 days in the presence of sodium formate and calcium propionate.

Items	Treatment ^1^			SEM	*p*-Value
	CON	SF	CAP		
Fermentation Characteristics					
pH	4.44 ^a^	4.11 ^b^	4.40 ^a^	0.051	0.0068
Ammonia-N, % of DM	0.237 ^a^	0.150 ^c^	0.185 ^b^	0.005	<0.0001
Lactic acid, % of DM	1.41 ^b^	2.57 ^a^	1.70 ^b^	0.15	0.0036
Acetic acid, % of DM	4.25 ^a^	0.30 ^c^	2.04 ^b^	0.057	<0.0001
Propionic acid, % of DM	ND	ND	2.2	-	-
Butyric acid, % of DM	2.81	ND	1.76	-	-
Chemical compositions					
DM, % of FW	25.75 ^c^	27.55 ^a^	26.16 ^b^	0.077	<0.0001
WSC, % of DM	0.21 ^c^	0.43 ^a^	0.29 ^b^	0.017	0.0002
CP, % of DM	27.80	27.17	27.63	0.39	0.53
NDF, of DM	62.91 ^c^	66.79 ^a^	64.06 ^b^	0.30	0.0003
ADF, % of DM	20.50	20.20	20.45	0.56	0.92
In situ effective degradability					
ISDMD ^2^ % of DM	47.30 ^c^	51.75 ^a^	49.33 ^b^	0.56	0.0039
ISNDFD ^2^ % of NDF	39.37 ^c^	44.11 ^a^	41.30 ^b^	0.51	0.0017
ISCPD ^2^ % of CP	54.16 ^b^	57.16 ^a^	55.52 ^b^	0.46	0.0111

ADF, acid detergent fiber; CP, crude protein; DM, dry matter; FW, fresh weight; ND, not detected; NDF, neutral detergent fiber; WSC, water-soluble carbohydrates; SEM, standard error of the mean. ^a–c^ Means within a row with different superscripts differ from each other (*p* < 0.05).^1^ Control = untreated feed; SF = sodium formate (3 g/kg FW); CAP = calcium propionate (3 g/kg FW). ^2^ ISDMD = in situ effective DM degradability; ISNDFD = in situ effective NDF degradability; ISCPD = in situ effective CP degradability.

**Table 3 animals-10-01608-t003:** Relative abundance (%) of 10 most predominant genera isolated from wet brewers grains treated without (control) or with sodium formate, or calcium propionate and ensiled for 20 days.

Genus	Treatment ^1^			SEM	*p*-Value
	CON	SF	CAP		
*Lactobacillus*	40.2 ^b^	84.6 ^a^	51.9 ^b^	3.61	<0.0001
*Clostridium sensu stricto 11*	36.7 ^a^	0.03 ^b^	18.1 ^a,b^	5.48	<0.01
*Clostridium sensu stricto 12*	6.74 ^a^	0.12 ^b^	9.53 ^a^	1.17	<0.01
*Prevotella 7*	3.58 ^a,b^	0.06 ^b^	7.16 ^a^	1.26	0.02
*Clostridium sensu stricto 1*	7.76 ^a^	0.00 ^b^	0.40 ^b^	0.80	<0.01
*Bacillus*	0.81 ^b^	4.53 ^a^	2.06 ^b^	0.46	<0.01
*Weissella*	0.36 ^b^	4.07 ^a^	0.46 ^b^	0.74	0.02
*Bromus tectorum*	1.57	1.00	1.46	0.17	0.12
*Bifidobacterium*	0.20 ^b^	0.21 ^b^	3.54 ^a^	0.69	0.02
*Clostridium sensu stricto 16*	0.11 ^b^	0.02 ^b^	3.11 ^a^	0.25	<0.01

^a,b^ Means within a row with different superscripts differ from each other (*p* ≤ 0.05).^1^ CON = untreated feed; SF = sodium formate (3 g/kg FW); CAP = calcium propionate (3 g/kg FW).

**Table 4 animals-10-01608-t004:** Pearson correlation coefficients between the abundance of bacterial genera and fermentation indices of wet brewers grains ensiled for 20 days.

Genus	pH		Lactic Acid (% of DM)		Acetic Acid (% of DM)		Propionic Acid (% of DM)		Butyric Acid (% of DM)		Ammonia-N (% of DM)	
	r	*p*	r	*p*	r	*p*	r	*p*	r	*p*	r	*p*
*Lactobacillus*	−0.73	**0.025**	0.77	**0.016**	−0.80	**0.010**	−0.11	0.780	−0.83	**0.006**	−0.85	**0.003**
*Clostridium sensu stricto 11*	0.73	**0.025**	−0.80	**0.010**	0.85	**0.004**	0.07	0.859	0.90	**0.001**	0.85	**0.004**
*Clostridium sensu stricto 12*	0.92	**0.001**	−0.43	0.244	0.48	0.188	0.64	0.061	0.56	0.117	0.59	0.092
*Prevotella 7*	0.65	0.058	−0.48	0.188	0.57	0.112	0.66	0.051	0.51	0.162	0.56	0.116
*Clostridium sensu stricto 1*	0.75	**0.019**	−0.76	**0.017**	0.84	**0.005**	0.02	0.960	0.87	**0.002**	0.90	**0.001**
*Bacillus*	−0.85	**0.004**	0.68	**0.042**	−0.87	**0.003**	−0.07	0.859	−0.92	**0.001**	−0.93	**0.0003**
*Weissella*	−0.92	**0.001**	0.55	0.125	−0.77	**0.016**	−0.29	0.454	−0.81	**0.008**	−0.83	**0.006**
*Bromus tectorum*	0.30	0.433	−0.65	0.058	0.63	0.067	0.23	0.556	0.63	0.071	0.34	0.366
*Bifidobacterium*	0.27	0.488	−0.02	0.966	0.05	0.898	0.78	**0.013**	0.00	1	−0.01	0.983
*Clostridium sensu stricto 16*	0.62	0.077	−0.50	0.171	0.52	0.154	0.82	**0.007**	0.46	0.215	0.46	0.213

The *p*-value corresponds to the significance of the bolded values.

## Data Availability

The Illumina sequencing raw data for our samples have been deposited in the NCBI Sequence Read Archive (SRA) under accession number: PRJNA661619.

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
