# Peer review of "Effects of Sodium Formate and Calcium Propionate Additives on the Fermentation Quality and Microbial Community of Wet Brewers Grains after Short-Term Storage"

_animals, 2020, doi:10.3390/ani10091608_

Round 1
Reviewer 1 Report
Manuscript ID: animals-896144
Title: Effects of Sodium Formate and Calcium Propionate Additives on the Fermentation Quality and Microbial Community of Wet Brewers Grains after Short-Term Storage
Authors: Jingyi Lv, Guangning Zhang, Chao Zhao, Xinpeng Fang, Yonggen Zhang, Yang Li
The manuscript addresses the problem of short-term preservation of wet brewers grains for use as a feed additive for cattle and dairy cows. The authors treated wet brewers grains with sodium formate and calcium propionate and examined their effect on fermentation products and microbial community after short-time (20 days) storage. They concluded that sodium formate is a promising preservative additive that increases contribution of lactic acid bacteria, especially Lactobacillus and improves the quality of wet brewers grains.
It is a short, well-done study.
Critical remarks:
- A list of abbreviations is required. The manuscript includes a lot of abbreviations and sometimes is difficult to read.
- Lines 96-99, the information that the wet brewers grains were ensiled in the afternoon should be removed. It is also a bit confusing when you write that the wet brewers grains were ensiled with two organic salts: (1) no additive, (2) sodium formate, (3) calcium propionate. Two organic salts were sodium formate and calcium propionate, the third group was untreated control. Of course, I understand what you mean but it sounds strange. The same is in the abstract.
- As for the in situ rumen degradation experiment. Was the bag 20x30 cm (one bag?) put to the rumen and every 4 hours the samples were collected for analyses?
- When you analysed the amplicon sequences what data base was used to assign taxonomy?
- Table 2 – is it possible to add propionic and butyric acids concentrations, instead of ND?
- The data in Table 2 show that the differences between treated and untreated wet brewers grains were rather small, however, statistically significant. Could you discuss significance of the difference between pH 4.1 and 4.4 comparing to other studies. Is the difference Δ=0.3 really important in the preservation process?
- Could you discuss significance of the differences between parameters ISNDFD, % of NDF or ISCPD, % of CP for the digestion process (39.4 vs 44.1 and 54.2 vs 57.2, respectively). Are the differences Δ ~3-5% really relevant for the digestion process in the rumen?
-
The raw DNA sequences generated in the reviewed study shoud be deposited in NCBI databases and the accession number should be given in the text. I can find any information about it.
Author Response
Comments from the editors and reviewers:
-Reviewer 1
The manuscript addresses the problem of short-term preservation of wet brewers grains for use as a feed additive for cattle and dairy cows. The authors treated wet brewers grains with sodium formate and calcium propionate and examined their effect on fermentation products and microbial community after short-time (20 days) storage. They concluded that sodium formate is a promising preservative additive that increases contribution of lactic acid bacteria, especially Lactobacillus and improves the quality of wet brewers grains.
It is a short, well-done study.
Critical remarks:
- A list of abbreviations is required. The manuscript includes a lot of abbreviations and sometimes is difficult to read.
Response: Thank you for your suggestion. According to the requirements of the magazine, abbreviations should be defined in parentheses the first time they appear in the abstract, main text, and in figure or table captions and used consistently thereafter. We carefully checked the new manuscript and confirmed that the abbreviations in the manuscript have been marked where they first appeared.
- Lines 96-99, the information that the wet brewers grains were ensiled in the afternoon should be removed. It is also a bit confusing when you write that the wet brewers grains were ensiled with two organic salts: (1) no additive, (2) sodium formate, (3) calcium propionate. Two organic salts were sodium formate and calcium propionate, the third group was untreated control. Of course, I understand what you mean but it sounds strange. The same is in the abstract.
Response: Thank you for your suggestion. In new manuscript, the information that the wet brewers grains were ensiled in the afternoon had been deleted.
We have modified the relevant content about treatment in Materials and methods. In new manuscript (Line 97), “Fresh WBG were delivered in the morning and ensiled in the afternoon of the same day with two organic salts: …” was changed into “Fresh WBG was treated with …” In addition, we think that the relevant content about treatment in Abstract meets your requirements (In the laboratory environment, fresh WBG was ensiled with (1) no additive (CON), (2) sodium formate (SF, 3 g/kg fresh weight), and (3) calcium propionate (CAP, 3 g/kg fresh weight) for 20 days.). Hope you can agree to our changes.
- As for the in situ rumen degradation experiment. Was the bag 20x30 cm (one bag?) put to the rumen and every 4 hours the samples were collected for analyses?
Response: Thank you for your suggestion. The bags (20x30 cm) in original manuscript was used as a fermentation bag for WBG silage. In the in-situ rumen degradation experiment, we do not collect samples every 4 hours for determination, but refer to the methods of Nuez-Ortín and Yu to determine in-situ rumen degradation at 48, 36, 24, 16, 12, 8, 4, and 0 hours, respectively. In this reference, the author described the method for determining rumen degradation parameters in a very detailed manner. Therefore, in our manuscript, we did not describe the method too much. Hope you understand. If you think it is necessary to add specific methods for determining rumen degradation parameters, we will add them in subsequent modifications.
Nuez-Ortín, W. G.;Yu, P. Estimation of ruminal and intestinal digestion profiles, hourly effective degradation ratio and potential N to energy synchronization of co-products from bioethanol processing. J. Sci. Food Agr. 2010, 90, 2058-2067.
- When you analysed the amplicon sequences what data base was used to assign taxonomy?
Response: Thank you for your suggestion. When analyzing the amplicon sequences, we used the Greengenes database.
- Table 2 – is it possible to add propionic and butyric acids concentrations, instead of ND?
Response: Thank you for your suggestion. The concentrations of organic acids (lactic acid, acetic acid, propionic acid and butyric acid) was measured using high performance liquid chromatography (HPLC) in Table 2. The measurement results did not show the peak values of propionic acid and butyric acid. Therefore, we cannot provide specific concentrations, and can only express the result as undetected. Such an expression method is the same as in many documents (Wang et al., 2019; Yan et al., 2019; He et al., 2020). Please forgive us that we still want to maintain the inter-statement form (ND).
Wang C, He L, Xing Y, et al. Fermentation quality and microbial community of alfalfa and stylo silage mixed with Moringa oleifera leaves[J]. Bioresource Technology, 2019, 284:240-247.
Yan Y, Li X, Guan H, et al. Microbial community and fermentation characteristic of Italian ryegrass silage prepared with corn stover and lactic acid bacteria[J]. Bioresource Technology, 2019, 279: 166-173.
He LW, Wang C, Xing YQ, et al. Ensiling characteristics, proteolysis and bacterial community of highmoisture corn stalk and stylo silage prepared with Bauhinia variegate flower[J]. Bioresource Technology, 2020, 296: 122336.
- The data in Table 2 show that the differences between treated and untreated wet brewers grains were rather small, however, statistically significant. Could you discuss significance of the difference between pH 4.1 and 4.4 comparing to other studies. Is the difference Δ=0.3 really important in the preservation process?
Response: Thank you for your suggestion. We can understand your opinion. The pH of the silage is a traditional and effective indicator for evaluating fermented feed. In this experiment, although the pH difference is small, the quality of the silage is ensured only when the pH of the silage is below 4.2. The pH value of the SF group after fermentation was lower than 4.2, while the pH value of the control group and the CP group after fermentation was higher than 4.2. Although the difference is only 0.3, this difference Δ 0.3 is very important for the fermentation process.
In addition, other studies (Wen et al., 2017) have also proved that formic acid can reduce the pH of alfalfa silage to 4.22, while calcium propionate can only maintain the pH of alfalfa silage at 4.48. It is concluded that after 30 days of ensiling, the addition of calcium propionate did not reduce silage pH to the level that ensure good conservation of alfalfa silage. We also made corresponding discussions in the Discussion section.
(In new manuscript (Lines 311-314), CAP plays a role in the preservation of fresh alfalfa; however, we found that CAP did not effectively reduce the pH of the fermented feed according to the literature data. This feature may explain why the CAP group did not have good fermentation properties in our experiment.)
Wen, A. Y.;Yuan, X. J.;Wang, J.;Desta, S. T.;Shao, T. Effects of four short-chain fatty acids or salts on dynamics of fermentation and microbial characteristics of alfalfa silage. Anim. Feed Sci. Tech. 2017, 223, 141-148.
- Could you discuss significance of the differences between parameters ISNDFD, % of NDF or ISCPD, % of CP for the digestion process (39.4 vs 44.1 and 54.2 vs 57.2, respectively). Are the differences Δ ~3-5% really relevant for the digestion process in the rumen?
Response: Thank you for your suggestion. ISNDFD and ISCPD are the effective degradation rates of NDF and CP in the rumen of WBG, respectively. In this experiment, we used SAS 9.4 software to process the ISNDFD and ISCPD data, and found that WBG has a significant difference in the effective degradation rate. We consulted some literature (Chen et al., 2019), and found that the difference between in vitro dry matter digestibility and in vitro neutral detergent fibre digestibility is small. However, such a difference of 3-5% can significantly change the rumen degradation process and the subsequent total-tract apparent digestibility (Hao et al., 2017). And, in our manuscript, we also discussed the importance of ISNDFD and ISCPD changes.
(In new manuscript (Lines 348-356), In this study, the addition of SF and CAP increased the in situ effective degradability of DM and CP apparently due to a reduction in the losses of the WBG silages treated with additives and consequently provide more available substrates for microbial degradation in the rumen [37]. Addition of SF increased the in situ effective degradability of NDF compared with that in the control and CAP groups; this result may be due to better fermentation effects of SF which reduce the loss of hemicellulose and other components in NDF caused by undesirable microorganisms; thus, the content of hemicellulose in WBG of the SF group was higher than that in the control and CAP groups. Hemicellulose is an easier fermentable fiber, and its content will directly influence the rate and extent of NDF degradation in the rumen [10].)
Chen, L.;Yuan, X. J.;Li, J. F.;Dong, Z. H.;Wang, S. R.;Guo, G.;Shao, T. Effects of applying lactic acid bacteria and propionic acid on fermentation quality, aerobic stability and in vitro gas production of forage-based total mixed ration silage in Tibet. Anim. Prod. Sci. 2019, 59, 376.
Hao X Y , Gao H , Wang X Y , et al. Replacing alfalfa hay with dry corn gluten feed and Chinese wild rye grass: Effects on rumen fermentation, rumen microbial protein synthesis, and lactation performance in lactating dairy cows[J]. Journal of Dairy ence, 2017, 100(4):2672.
- The raw DNA sequences generated in the reviewed study shoud be deposited in NCBI databases and the accession number should be given in the text. I can find any information about it.
Response: Thank you for your suggestion. We have uploaded the original 16sRNA data in NCBI and got the accession number. In new manuscript (Lines 428-429), the relevant content had been added. The new content are: “Data Availability: The Illumina sequencing raw data for our samples have been deposited in the NCBI Sequence Read Archive (SRA) under accession number: PRJNA661619.”

Reviewer 2 Report
Dear Authors
In my opinion the paper is well written, clear and complete.
Just some suggestion: why the ensilage process was stopped at 20 d? are they enough. Other papers report at least 35 d (i.e. Johnson et al 2005 Journal of Applied Microbiology, 98: 106-113).
P4L185-188: in the text the same data indicated in the table 1 are reported, as well as table 4 and text in 3.4 paragraph.
Why microrganisms are reported as % DM and not as unit forming colony (UFC)?
Haven't you measured the buffer capacity? could the high protein content have affected the pH?
In the conclusion a practical use of the additives proposed could be added.
Author Response
-Reviewer 2
Comments and Suggestions for Authors
Dear Authors
In my opinion the paper is well written, clear and complete.
- Just some suggestion: why the ensilage process was stopped at 20 d? are they enough. Other papers report at least 35 d (i.e. Johnson et al 2005 Journal of Applied Microbiology, 98: 106-113).
Response: First of all, thank you very much for your valuable comments. We have carefully read the literature you provided. Before the start of this experiment, we understood the problems faced by wet brewer's grains as the main feed ingredient in the pasture. In addition, we also referred to some articles (Allen et al., 1975a; Allen et al., 1975b; Hatungimana et al., 2019) on the preservation of wet brewer's grains before the study, and found that the short-term effective storage of wet brewer's grains can meet the requirements of the ranch for the use of wet brewer's grains. The short-term fermentation days of wet brewer's grains reported in these articles are 14 days, 18 days, or 28 days. Therefore, considering the use and storage time of the raw materials in the pasture and the fermentation effect in the literature (The results of the 18-day study suggested that a silage additive would assist in satisfactory ensiling of wet brewers' grains. (Allen et al., 1975b) ), we have determined that the fermentation time of this experiment is 20 days, and we also think that it can have a good fermentation effect. And, in future research, we will extend the fermentation time and explore the effect of long-term fermentation of wet brewers' grains with organic acid salts. We hope our answer will satisfy you, and We hope you can agree with our experimental design. Thank you again for your valuable comments.
Allen, W.R., Stevenson, K.R. Buchanan-Smith. J. 1975a. Influence of additives on short-term preservation of wet brewers' grain stored in uncovered piles. Can. J. Anim. Sci. 55: 609-618.
Allen, W.R., Stevenson, K.R. 1975b. Influence of additives on the ensiling process of wet brewers' grain. Can. J. Anim. Sci. 55: 391-402.
Hatungimana, E., Erickson, P.S. 2019. Effects of storage of wet brewers grains treated with salt or a commercially available preservative on the prevention of spoilage, in vitro and in situ dry matter digestibility, and intestinal protein digestibility. Applied Anim. Sci. 35:464–475.
- P4L185-188: in the text the same data indicated in the table 1 are reported, as well as table 4 and text in 3.4 paragraph.
Response: Thank you for your suggestion. 3.1 and 3.4 paragraph is the description of the results in Table 1 and Table 4. In other articles of Animals (Liu et al., 2019), the Results section (a description of tables) is necessary to help readers understand the table. In our manuscript, Table 1 shows chemical compositions and microbial populations of wet brewers grains before ensiling, and Table 4 shows pearson correlation coefficients between the abundance of bacterial genera and fermentation indices of wet brewers grains. The data in the two tables are relatively intuitive and cannot carry out a deeper level of comparative analysis, so the results section will be duplicated with the data in the table. We hope the reviewers understand and thank you again for your valuable comments. If you have any comments, please feel free to contact us, we will make further changes based on your comments.
Liu KY, Li Y, Luo GB, et al. Relations of ruminal fermentation parameters and microbial matters to odd- and branched-chain fatty acids in rumen fluid of dairy cows at different milk stages[J]. Animals, 2019, 9(12):1019.
- Why microrganisms are reported as % DM and not as unit forming colony (UFC)?
Response: Thank you for your suggestion. In original manuscript (Table 1), the unit of microorganisms is cfu (colony forming units). We checked the new manuscript and confirmed that the unit of microorganisms is not % DM.Thank you again for your valuable comments.
- Haven't you measured the buffer capacity? could the high protein content have affected the pH?
Response: Thank you for your suggestion. Unfortunately, in this experiment, we did not test the buffer capacity of wet brewer's grains. For feed protein, SCP is the main factor affecting feed buffering capacity and pH. The ratio of SCP to CP in wet brewer's grains is very low (Measured (not shown); Qu, et al., 2010; Li Qian, 207), which is much lower than that of alfalfa and other feeds with strong buffering capacity. In addition, we believe that the influence of additives on the pH of the fermented feed is much greater than that of the protein in wet brewer's grains. Therefore, the protein of wet brewer's grains has little effect on pH. So we did not test the buffer capacity of wet brewer's grains.
Qu YL, Wu JH, Li T. Use of cornell net carbohydrate and protein system for evaluation of nutrient value of feeds to dairy cattle in the northeast agricultural region of china[J]. Chinese Journal of Animal Nutrition, 2010, 22(1):201-206.
Li Qian. Study on nutrients composition and ruminal fermentation characteristics of difference types distillers’ grains [M]. Ya’an, Sichuan, P.R. China
- In the conclusion a practical use of the additives proposed could be added.
Response: Thank you for your suggestion. The new contents had been added in new manuscript (Lines 417-418). The contents are: “In production, the addition of SF can prolong the storage time of WBG, thereby improving the operability of using of WBG on dairy farm.”
